# National parochialism is ubiquitous across 42 nations around the world

Angelo Romano [1,2✉], Matthias Sutter [1,3,4], James H. Liu[5], Toshio Yamagishi[6] & Daniel Balliet [7,8]

Cooperation within and across borders is of paramount importance for the provision of public goods. Parochialism – the tendency to cooperate more with ingroup than outgroup members – limits contributions to global public goods. National parochialism (i.e., greater cooperation among members of the same nation) could vary across nations and has been hypothesized to be associated with rule of law, exposure to world religions, relational mobility and pathogen stress. We conduct an experiment in participants from 42 nations ($N = 18,411$), and observe cooperation in a prisoner's dilemma with ingroup, outgroup, and unidentified partners. We observe that national parochialism is a ubiquitous phenomenon: it is present to a similar degree across the nations studied here, is independent of cultural distance, and occurs both when decisions are private or public. These findings inform existing theories of parochialism and suggest it may be an obstacle to the provision of global public goods.

[1] Experimental Economics Group, Max Planck Institute for Research on Collective Goods, Bonn, Germany. [2] Leiden University, Social, Economic and Organizational Psychology, Leiden, The Netherlands. [3] Department of Economics, University of Cologne, Cologne, Germany. [4] Department of Public Finance, University of Innsbruck, Innsbruck, Austria. [5] School of Psychology, Massey University, Auckland, New Zealand. [6] Graduate School of International Corporate Straegy, Hitotsubashi University, Tokyo, Japan. [7] Department of Experimental and Applied Psychology, Vrije Universiteit Amsterdam, Amsterdam, The Netherlands. [8] Institute of Brain and Behavior (IBBA), Amsterdam, The Netherlands. ✉email: a.romano@fsw.leidenuniv.nl

Humans possess a remarkable ability to cooperate to create public goods that are shared among genetically unrelated individuals[1–3]. Several theories predict that this cooperation may be parochial, i.e., people cooperate more with members of their own group compared to members of other groups[4–9]. In an increasingly interdependent world, national parochialism may substantially impede the provision of public goods across nations, such as containing a pandemic disease (e.g., Covid-19[10]), environmental conservation, or the creation of multi-national institutions (e.g., the United Nations or the European Union). While national parochialism has been hypothesized to be a widespread phenomenon, it has also been argued that its extent should vary substantially across ecologies, nations, and cultures[11–17]. Given the potential downside of national parochialism for cooperation across nations, it is important for both theory and policy to understand better the ubiquity, variability, and determinants of national parochialism across nations.

Several evolutionary mechanisms have been proposed to explain parochialism[18,19]. One possibility is that people expect more cooperation from ingroup members, are more concerned about their reputation with ingroup vs. outgroup members, and cooperate with ingroup members to acquire indirect benefits (e.g., a positive reputation) and avoid exclusion from the group[4,19]. If true, national parochialism should be pervasive across nations and only occur when cooperative behavior is observed by ingroup members.

Alternatively, cultural evolution theory proposes that culture shaped modern large-scale cooperation—and especially parochialism—through norms and institutions that have emerged over the course of human history[11,12]. Norms and institutions may have affected the relevant categories for discrimination, such as the extent of national parochialism around the world[16]. Specifically, institutions that provide a safety net for interactions among strangers and across group boundaries (e.g., rule of law or government effectiveness) would promote mutually beneficial social exchange and reduce national parochialism[17]. Modern world religions are also cultural institutions that are hypothesized to promote and enforce norms of cooperation between groups, thereby reducing national parochialism[10,20]. From these perspectives, nations are expected to differ in their extent of national parochialism, and cultural differences in social institutions like rule of law and modern world religions should co-vary with national parochialism.

To date, empirical evidence testing these different models of national parochialism is limited to self-report measures, or experiments conducted in a limited set of nations, including mostly Western, Educated, Industrial, Rich, and Democratic (WEIRD) nations[4,21–24]. Here, we report an experiment run in December 2018, involving a sample of 18,411 participants representative for age, gender, and income, from 42 nations (Table 1), to test pre-registered hypotheses (https://osf.io/gnxv2/) about the prevalence and variation of national parochialism across nations. Some variables from this project have also been used in two other papers. In particular, one paper examined the relation between political ideology, cooperation, and national parochialism[25], while another paper has examined how cooperation and trust relate with responses to the COVID-19 pandemic[26]. Participants were recruited online through the Harris panel (including members of its third-party providers, see "Methods"). Altogether, these 42 nations account for ~73% of the world population[27].

Participants completed an online experiment, and were asked to make 12 independent decisions in a prisoner's dilemma game (PD), each decision with a different partner and without feedback. In the PD, participants were endowed with 10 Monetary Units (MUs) and could decide how many of them to keep for themselves and how many to give to their partner. They understood that each MU given to their partner was doubled (while those kept were not doubled), and that their partner also had the option to give any amount to them, and that this amount too would be doubled. To make decisions comparable across nations, participants were informed that each MU was worth 2.5 min of the average hourly wage in their nation.

In each decision, participants were randomly assigned to interact with a person from their own nation (ingroup treatment), a person randomly selected from one of 16 possible other nations (outgroup treatment), or a person without any information provided about their nationality (stranger treatment). Across the 12 decisions, there were two blocks of six decisions each that varied whether or not one's decision would become public or stay private (note that each block contained two interactions each with an ingroup member, outgroup member, and stranger). In the public block, participants provided a pseudonym, and were told that their choices would be published online using their pseudonym. In the private block, participants knew that their choices would not be published online. Importantly, previous work found that people respond to cues of whether indirect reciprocity is possible (e.g., being observed) even when actual opportunities for indirect reciprocity are not present[24]. Moreover, past research used this method to manipulate observability and reputational consequences of behavior and found this to increase cooperation[28,29], and we replicated this finding in a pilot study using our current experimental design (see "Methods"). After each decision, we asked participants about their expectation for their partner's level of cooperation.

In the present work, we find that national parochialism is a pervasive phenomenon that occurs around the world and with little variation across nations. We do not find support for the prediction that national parochialism only occurs in public (vs. private) situations. Rather, public situations result in greater cooperation compared to private situations, regardless of partner nationality. Relative to national parochialism, we find greater variation across nations in how people cooperate with strangers (i.e., impersonal cooperation), independent from partner nationality. Cross-national variation in impersonal cooperation is associated with social norms, institutions, and ecological conditions. In sum, our findings highlight the ubiquity of national parochialism across 42 nations around the world, and how there is a relatively greater amount of variation in impersonal cooperation across these nations.

## Results

Across all 42 nations, participants made decisions about hypothetical MUs. A meta-analysis of 212 experiments found that parochialism occurs to an equal extent when decisions involve actual vs. hypothetical incentives[4,24]. Furthermore, a recent study manipulated actual vs. hypothetical incentives in a trust game, both in the UK and South Korea, and found equal amounts of national parochialism when people were paid or not for their decisions[24]. Nonetheless, we tested whether using hypothetical choices could have influenced the participants' decisions. In three nations (i.e., Brazil ($n = 832$), India ($n = 834$) and Poland ($n = 776$)), we randomly assigned participants to either an incentives treatment (with monetary payments) or hypothetical incentives treatment. Cooperation was assessed by the amount of MUs people sent to their partner. National parochialism was indexed by the (positive) difference in cooperation with ingroup members vs. outgroup members and unidentified strangers (see Supplementary Information, section 1.1.8 for details on the models). Consistent with previous research[4,24], we found no difference between incentivized and hypothetical treatments in cooperation

**Table 1 Summary of descriptives.**

| Nation | N | Language of the survey | $M_{age}$ (SD) | %Females | %Coop | Cohen's d |
|---|---|---|---|---|---|---|
| Argentina | 387 | Spanish | 35.02 (11.09) | 51.42 | 43.82 | 0.20 |
| Australia | 383 | English | 45.41 (13.40) | 57.96 | 42.37 | 0.34 |
| Bolivia | 391 | Spanish | 30.40 (9.16) | 46.80 | 49.46 | 0.15 |
| Brazil | 832* | Portuguese | 34.50 (10.83) | 61.66 | 38.68 | 0.11 |
| Canada | 379 | English | 44.11 (13.32) | 55.94 | 44.83 | 0.40 |
| China | 393 | Chinese | 30.61 (8.36) | 53.18 | 35.83 | 0.38 |
| Colombia | 399 | Spanish | 34.45 (11.81) | 53.88 | 43.71 | 0.17 |
| Egypt | 408 | Arabic | 29.44 (8.50) | 45.59 | 41.28 | 0.24 |
| Finland | 388 | Finnish | 40.95 (13.27) | 54.64 | 45.08 | 0.30 |
| Germany | 391 | German | 44.90 (12.73) | 51.92 | 39.88 | 0.20 |
| Greece | 396 | Greek | 40.19 (11.66) | 54.55 | 37.28 | 0.38 |
| Hong Kong | 390 | Complex Chinese | 35.67 (11.00) | 51.79 | 38.32 | 0.06 |
| Hungary | 391 | Hungarian | 41.27 (12.95) | 56.78 | 46.12 | 0.21 |
| India | 834* | Hindi, English | 33.37 (11.09) | 45.92 | 36.92 | 0.20 |
| Indonesia | 384 | Indonesian | 34.00 (10.12) | 49.74 | 35.06 | 0.20 |
| Italy | 675 | Italian | 41.33 (12.65) | 57.19 | 39.88 | 0.18 |
| Japan | 393 | Japanese | 48.06 (11.82) | 52.42 | 33.76 | 0.16 |
| Kenya | 383 | English | 30.48 (9.05) | 49.61 | 38.93 | 0.17 |
| Malaysia | 404 | Malay, English, Chinese | 33.85 (10.30) | 50.74 | 38.64 | 0.26 |
| Mexico | 408 | Spanish | 33.90 (11.52) | 56.37 | 40.65 | 0.19 |
| Morocco | 394 | Arabic | 31.85 (9.68) | 49.49 | 40.25 | 0.25 |
| Netherlands | 653 | Dutch | 46.45 (13.36) | 55.28 | 42.55 | 0.29 |
| New Zealand | 386 | English | 41.89 (13.04) | 59.33 | 47.27 | 0.31 |
| Nigeria | 395 | English | 30.93 (9.58) | 44.30 | 40.75 | 0.06 |
| Pakistan | 388 | Urdu | 28.86 (8.85) | 24.48 | 41.43 | 0.11 |
| Panama | 397 | Spanish | 31.55 (10.17) | 53.65 | 45.60 | 0.20 |
| Peru' | 393 | Spanish | 34.12 (10.56) | 58.78 | 44.34 | 0.05 |
| Philippines | 384 | Filipino | 34.56 (11.04) | 55.73 | 36.83 | 0.26 |
| Poland | 776* | Polish | 39.37 (13.07) | 53.22 | 42.21 | 0.11 |
| Portugal | 448 | Portuguese | 37.98 (11.64) | 55.58 | 40.69 | 0.25 |
| Russia | 387 | Russian | 39.76 (11.09) | 51.16 | 39.93 | 0.32 |
| Serbia | 390 | Serbian | 37.60 (11.88) | 55.13 | 43.48 | 0.24 |
| Singapore | 384 | English | 39.58 (11.99) | 51.82 | 36.79 | 0.26 |
| South Africa | 390 | English | 35.24 (11.48) | 52.31 | 41.29 | 0.14 |
| South Korea | 379 | Korean | 40.90 (11.47) | 57.78 | 41.49 | 0.28 |
| Spain | 389 | Spanish | 41.16 (11.52) | 56.81 | 43.21 | 0.15 |
| Sweden | 392 | Swedish | 43.15 (12.59) | 53.57 | 45.16 | 0.15 |
| Taiwan | 392 | Complex Chinese | 36.31 (10.62) | 55.36 | 43.22 | 0.22 |
| Turkey | 414 | Turkish | 33.45 (9.97) | 44.20 | 42.85 | 0.35 |
| UK | 433 | English | 42.54 (12.78) | 56.12 | 42.91 | 0.17 |
| United States | 378 | English | 42.44 (13.30) | 60.05 | 48.27 | 0.23 |
| Venezuela | 435 | Spanish | 35.36 (11.53) | 47.59 | 47.47 | 0.44 |
| Total | 18,411 | | 37.40 (12.52) | 52.77 | 41.49 | 0.22 |

Nations, sample sizes, %females, mean age, language of the survey, average cooperation, Cohen's d = standardized mean difference in cooperation between ingroup vs. outgroup and strangers (positive = more cooperation with ingroup). Source data are provided as a Source Data file.
*Nation with the incentive treatment.

($b = -0.14$, $p = 0.12$) and national parochialism ($b = 0.01$, $p = 0.74$). Incentives also did not change the effect of observability on cooperation ($b = 0.04$, $p = 0.34$). Thus, based on a meta-analysis of the literature and on evidence from five nations where incentives have been manipulated (Brazil, India, Poland, South Korea, and UK), we conclude that the psychology of national parochialism does not respond differently to hypothetical vs. incentivized decisions. Therefore, we study national parochialism in the absence of incentives (except for the three nations mentioned above), which allowed us to maximize the sample size of participants and nations.

**National parochialism around the world.** Across all 42 nations, participants cooperated more when they knew that their partner was from the same nation, compared to when they knew that their partner was from another nation or a stranger (mixed-effects regression: $b = 0.29$, $p < 0.001$; see Supplementary Table 1–11 for a complete report of all results). In fact, there was statistically significant national parochialism in 39 out of 42 nations, and there was a non-significant, but still positive, amount of national parochialism in the remaining three nations (see Supplementary Figs. 1–3). In line with previous evidence[4], national parochialism was associated with a motivation to favor a person from one's own ingroup, rather than a motivation to derogate the outgroup (see Supplementary Table 11). We observed little variation in national parochialism across nations and cultures (see Fig. 1). Moreover, when we grouped nations into 10 cultural groups based on differences in history, politics, and values measured in the world values survey[30] we found significant national parochialism in each cultural grouping (see Supplementary Information, section 1.1.7). We further examined if cultural differences and similarities between each pair of nations were associated with differences in national parochialism. To do so, we used a composite proxy of

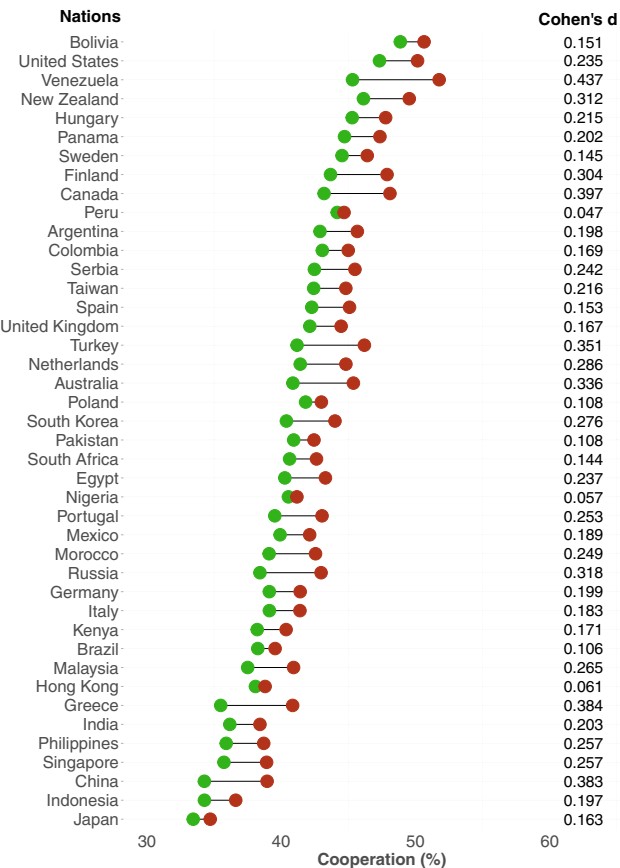

**Fig. 1 National parochialism across nations.** Cleveland dot plot showing the mean of cooperation (in percentage) with ingroup members (red dots) compared to outgroup members and unidentified strangers (green dots) across all the 12 decisions (including both the public and private treatments). Nations are sorted based on their average cooperation levels. The right-side column reports the estimated standardized mean difference (Cohen's $d_c$) of national parochialism in cooperation. Source data are provided as a Source Data file.

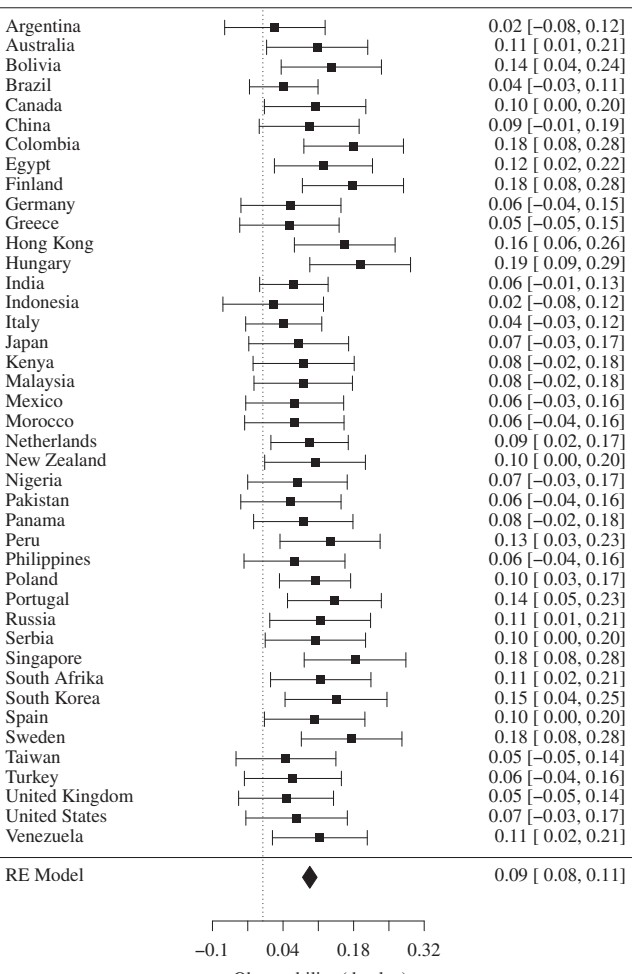

**Fig. 2 Effect sizes of observability across nations.** Forest plot displaying effect size of observability predicting cooperation. For each nation, we report estimated standardized mean difference (Cohen's $d_c$) and 95% confidence interval in the right side of the plot. A positive $d$ value indicates higher cooperation in the public treatment compared to the private treatment. The overall estimated population effect sizes are represented by the black diamonds, which correspond to the 95% confidence interval. Source data are provided as a Source Data file.

bilateral cultural distances[31] (see Supplementary Information, section 1.3). Cultural distance between nations did not predict differences in national parochialism ($b = -0.022$, $p = 0.70$), suggesting that nations which are culturally distant from each other display a similar degree of national parochialism.

While it is true that the level of cooperation was generally larger in the public block than the private block ($b = 0.12$, $p < 0.001$, see Fig. 2), national parochialism occurred in both the public and private block and was not significantly different across the two conditions ($p = 0.59$). This latter finding does not support the hypothesis that national parochialism is a strategy to acquire indirect benefits (e.g., a positive reputation) within a group[4]. Cooperation in the public block occurs regardless of whether the interaction partner is an ingroup member, an outgroup member, or an unidentified stranger. However, we found people expected more cooperation from ingroup members than outgroup members and strangers, and these positive expectations in turn were correlated with national parochialism (indirect effect: $b = 0.06$, $p < 0.001$).

We tested several hypotheses about the cultural and institutional factors that account for cross-national variation in national parochialism (see Supplementary Table 8). National parochialism did not vary across nations according to the quality of institutions (e.g., rule of law, $p = 0.56$; government effectiveness, $p = 0.77$), intensity of kinship norms and historical exposure to western

church[32] ($p = 0.52$, $p = 0.70$), and the prevalence of modern world religions (e.g., religiosity, $p = 0.44$, church attendance, $p = 0.11$). Finally, results also fail to support theories about how the ecology (e.g., pathogen stress[33], $p = 0.43$; relational mobility, $p = 0.66$) can affect national parochialism. All of this is evidence against models that assume meaningful and substantial differences in national parochialism across ecologies, nations, and cultures[17].

We observed higher variation in national parochialism within-nations ($SD_{within} = 1.25$) than between-nations ($SD_{between} = 0.13$). This variation is partially explained by gender differences and differences in the education level of participants (i.e., elementary school, middle school, high school, some college, bachelor's degree, graduate school or higher). We found that the relation between national parochialism and cooperation is stronger among men, compared to women ($b = 0.03$, $p = 0.04$; see Supplementary Table 4). Moreover, the relation between national parochialism and cooperation is weaker in people with higher, compared to lower, education ($b = -0.02$, $p = 0.02$; see Supplementary Table 5).

**Cooperation**. Overall, we found national parochialism is ubiquitous, suggesting that national parochialism is a stable and widespread feature of nations. Moreover, surprisingly, the degree of national parochialism varied little across nations that have, for example, different levels of wealth or political stability. This is not to say, however, that the level of cooperation itself (rather than the extent of national parochialism) does not differ across nations. In fact, we found that the general amount people were willing to give to their partner—as a form of impersonal cooperation—did vary across nations. This finding is consistent with the theory that cooperation and parochialism are two distinct behaviors that may have been shaped by different evolutionary mechanisms, and thus may not be associated with the same ecological, social and institutional factors that vary across nations[34]. Hence, although the focus of our paper and the pre-registered hypotheses concerned variation in national parochialism across nations, we continue with an exploratory analysis of ecological and cultural factors that can account for cross-national variation in impersonal cooperation.

We identified all ecological, social, and institutional factors (see Supplementary Table 12 for a summary of the indicators, sources, and year of measurement) that have been hypothesized to account for cross-national differences in cooperation, and then we ran OLS mixed-effects regressions where cooperation levels (i.e., the number of MUs transferred to the partner) were predicted by national-level indicators (see Supplementary Table 9). We also checked whether cultural distance between nations predict differences in cooperation, and so it does ($b = 0.59$, $p < 0.001$; see Supplementary Information, section 1.3). The greater the cultural distance, the larger the difference in cooperation. Moreover (see Supplementary Table 9), cooperation was higher in nations that are characterized by low historical prevalence of infectious diseases ($b = -0.65$, $p < 0.001$); by higher historical exposure to western churches[20] ($b = -0.53$, $p = 0.003$), by more egalitarian values (power distance, $b = -0.49$, $p = 0.006$; hierarchy[35], $b = -0.43$, $p = 0.04$); by informal and tolerant social norms (indulgence, $b = 0.49$, $p = 0.007$; tightness-looseness[36], $b = 0.40$, $p = 0.06$); by individualistic and self-expression values ($b = 0.39$, $p = 0.03$); and by more flexible and fluid social relations (i.e., national relational mobility, $b = 0.64$, $p = 0.009$). In our study, participants completed a measure of their perceived relational mobility[13], and these data replicated our findings at the individual-level: people who perceived their environment to have more opportunities to establish new relationships with strangers were generally more cooperative with both ingroup and outgroup members ($b = 0.35$, $p < 0.001$).

## Discussion

In conclusion, we report an experiment testing hypotheses on the prevalence and variation of national parochialism that includes a total of 42 nations. We found national parochialism is a pervasive phenomenon and it occurs around the world with very little variation across nations and cultures. Our findings failed to support prominent hypotheses predicting substantial variation in national parochialism around the world[4,13,14]. Rather, national parochialism seems to be a ubiquitous behavior across modern nations, a finding that is in line with an indirect reciprocity perspective, and in general with theories which hypothesize the pervasiveness of ingroup favoritism in humans[4,15,37,38]. However, contrary to what was predicted by an indirect reciprocity perspective, we failed to observe that national parochialism only occurred in public (vs. private) situations.

These findings are in contrast to past evidence from less industrialized societies that shows higher between-nation variance in ingroup favoritism across nations[39–42]. These inconsistencies

may be either due to differences among the participants in these studies, to the different interdependent situations (prisoner's dilemma vs allocation game) used to investigate ingroup favoritism in the studies[4], or to other crucial differences in the study designs (e.g., a decision of allocating resources between an ingroup member and outgroup member, vs distribute resources between themselves and another person who is either an ingroup member or outgroup member). Investigating the universality of ingroup favoritism among diverse groups (e.g., tribes, nations) with a comparable design is a challenge for future research. Moreover, we found that national parochialism in our study was associated with motivation for ingroup favoritism, rather than outgroup derogation. However, the prisoner's dilemma may not represent an ideal setting to study a motivation to harm outgroup members. Future cross-cultural research on national parochialism may use paradigms with an option to inflict costs on the outgroup[43].

Our findings are important because many social, environmental, and economic challenges demand cooperation across nations—and managing the tendency to favor ingroup members provides a crux in successfully solving these challenges when they transcend borders. Therefore, it is important for our future to pursue research on the conditions that expand people's willingness to cooperate more beyond group boundaries[21].

Relative to national parochialism, we observed greater variation in how people cooperate with strangers across nations, independent from partner nationality. It is especially cooperation between strangers—with no prior or future interactions—that can enable nations to scale-up public goods (through impersonal cooperation;[1–3,15]). Impersonal cooperation may be independent from the bias to favor cooperation with ingroup, compared to outgroup, members[6,44]. In fact, several ecological, social, and institutional factors have been hypothesized to account for cross-national differences in impersonal cooperation[12–15]. Our data can provide a compelling test for these theories, which are usually tested in a limited set of nations or regions[45]. We found that variation in impersonal cooperation is associated with cross-national differences in institutions (history of exposure to western church), norms (egalitarian, more tolerant), values (individualistic) and social ecological conditions (low pathogen stress, flexible and fluid social relations).

A few limitations of the present research are worth noting. First, the use of within-subjects manipulations of the partner's nationality and the observability of choices could produce demand characteristics, such as guessing the purpose of the study and behaving congruent with the hypotheses. Also, we did not ask participants what they thought were the reasons for their decisions. We did not feel compelled to ask these questions because past research has found that individuals are not accurate in the interpretation of their own mental states or behavior[46,47]. That said, previous research did not find a difference in behavior between studies that used either a within-subjects design or between-subjects design to investigate the effect of parochialism or observability on cooperation[4]. Moreover, if our study created demand characteristics, then we should observe a difference in behavior between hypothetical and incentivized treatments, as incentives are hypothesized to decrease the effect of demand characteristics[48,49]. We observed the same results for both hypothetical and incentivized treatments (see Supplementary Information, section 1.1.4). Second, people may cooperate more with national ingroup members because these interactions involve less uncertainty—a possible alternative explanation for the ubiquity of national parochialism. However, we observed a similar amount of variability in a partner's expected cooperation within interactions with outgroup members and strangers, compared to ingroup members ($sd_{ingroup} = 2.66$, $sd_{outgroup/strangers} = 2.60$). This result suggests that the national parochialism observed in our study is not explained by uncertainty. Finally, we found small to medium effect

sizes for national parochialism. Importantly, the observed effect size of national parochialism in our study ($d = 0.22$) is in line with previous research on ingroup favoritism[4], and similar to effect sizes observed within the social sciences, in general[50]. Moreover, the magnitude of effect sizes is usually lower in larger samples[51], and even small effect sizes can have substantial accumulative societal impacts over time[50,52,53].

In conclusion, despite the fact that previous theories about cross-national variation in cooperation have largely focused on parochialism[17,54], we observed a similar degree of national parochialism around the world. Our data suggest that theories about the cultural evolution of cooperation may be more relevant in explaining differences in impersonal cooperation[54]. Indeed, these results provide evidence that culture and ecology could have possibly shaped the human ability to cooperate with strangers. Thus, identifying strategies to promote cooperation between strangers, across nations and national boundaries, may result in solutions to the provision of public goods and management of pressing societal challenges within and across nations.

## Methods

The research and procedure (including the informed consent, see Supplementary Information, section 1.5) were approved by the Massey University Human Ethics Committee, application number: 4000019960 and by the board for Ethical Questions in Science of the University of Innsbruck, application number 37/2018.

**Participants.** We recruited 18,411 participants from 42 nations (Argentina, Australia, Bolivia, Brazil, Canada, China, Colombia, Egypt, Finland, Germany, Greece, Hong Kong, Hungary, India, Indonesia, Italy, Japan, Kenya, Mexico, Malaysia, Morocco, Netherlands, New Zealand, Nigeria, Pakistan, Panama, Peru, Philippines, Poland, Portugal, Russia, Serbia, Singapore, South Afrika, South Korea, Spain, Sweden, Taiwan, Turkey, Venezuela, United Kingdom, and United States). In line with previous research[24,55], participants in this study were recruited through a partnership with Nielsen, a media polling company based in the USA (including members of its third party panel providers), that encompasses >10 million individuals. Participants were stratified by age, gender, and income. Participants were either invited by email, or could have access to the link through the panelist portals.

**Power analysis.** Our goal was to detect discrimination between ingroup and outgroup members/strangers. A recent meta-analyses found an effect size of $d = 0.27$ for the within-subjects difference between people's willingness to cooperate with an ingroup member, compared to an outgroup member[4]. An a priori-power analysis[56] suggests that to detect this effect size at statistical power $(1-\beta) = 0.95$ and $\alpha = 0.05$ requires a sample size of 150 people per nation. A sensitivity power analyses that consider a sample size of 400 participants and a 95% statistical power and 5% of probability error, reveals that we can detect very small effect sizes of discrimination ($d = 0.16$).

**Procedure and experimental design.** Hypotheses and design were preregistered at https://osf.io/68wds/. The design consisted of three counter-balanced within-subject treatments related to the nationality of the interacting partner (partner's nationality: Ingroup vs. Outgroup vs. Stranger) and two within-subjects counter-balanced treatments that varied whether the choice was either private or public. The data were collected through an online survey using the Qualtrics software. We wrote an English version of the survey. After that, we asked experts to translate the survey by back-translation or the committee method. The procedure of the experiment was the same across all nations. After giving their informed consent (see Supplementary Information, section 1.5), participants were asked to make 12 independent cooperation decisions, each with a different partner. There was no debriefing at the end of the study. However, the study did not involve any deception and, in the informed consent section (see Supplementary Information, section 1.5), participants were provided contacts to the researchers to request further information about the study and the results. Participants were asked to complete a measure of relational mobility. Participants also responded to several additional questionnaires, which are not related to this project.

**Partner's nationality.** In the Ingroup treatment participants had to decide to cooperate or defect by giving between 0 and 10 MU to a partner from the same nation. In the Outgroup treatment, participants had to decide upon their level of cooperation (from 0 to 10 MU) with a partner from one of a set of other 16 nations. Since participants made two outgroup decisions per each observability treatment, we split the outgroup treatment in two sets of outgroup (Outgroup 1 = Canada, Hong Kong, Hungary, Kenya, New Zealand, Panama, Sweden, Venezuela;

Outgroup 2: Australia, Colombia, Germany, India, Nigeria, Serbia, Singapore, United States). In the Stranger treatment, the nationality of the partner was not specified.

**Public vs. private choices.** In the public treatment, participants knew that their choice would be published on a website under a nickname. The nickname was decided by the participant at the beginning of the study and was a string of two letters and two numbers. There is no real possibility to be personally identified from the nickname by the researchers or other third parties. The manipulation had the goal to increase a perception of observability while still no real personal information was provided. In the private treatment, participants knew that their choice would not be published on any website. We posted the decisions made in the public treatment on https://what-did-people-do.com/. We conducted a pilot study to test whether this manipulation was successful in promoting cooperation in online settings. We recruited 369 participants via MTurk, and had them complete our study, including decisions in the public and private treatments. We found that there was significantly more cooperation in the public treatment ($M = 4.62$, $SD = 3.42$), compared to the private treatment ($M = 4.27$, $SD = 3.42$; $F(1, 367) = 21.09$, $p < 0.001$, $\eta^2_p = 0.054$).

**Incentives.** Participants in Brazil, India and Poland ($N ≈ 800$ per nation) were randomly allocated to a between-subjects treatment where cooperation decisions could result in real monetary outcomes or a treatment where cooperation decisions resulted in hypothetical outcomes. Participants were endowed with 10 monetary units (MU). Then, they were informed that each MU corresponded to 2.5 min average wage in each nation. Information of wage in each nation were retrieved at https://tradingeconomics.com/country-list/wages. Participants were paid for one of the decisions in the incentive treatments. Results of the interaction between incentives, observability, and discrimination can be found in the Supplementary Information, section 1.1.4.

**Relational mobility.** Participants responded to what extent they agreed with seven statements about the relational mobility of people in their nation[13]. Specifically, they were asked to state how well seven statements described the people in the nation where they live. An example for such a statement is "It is common for these people to have a conversation with someone they have never met before" (1 = "Strongly Disagree", 7 = "Strongly Agree"). Higher scores indicate that people perceive their nation to promote open and flexible social relations.

**Attention check and proxy for quick understanding.** To assure quality of the data, we added a question where participants were asked to not respond. We excluded participants that did not pass this attention check in the main analysis. However, the results do not change when considering all participants (see Supplementary Information, section 1.1.2). We also had a proxy of understanding the prisoner's dilemma. As a comprehension question, we asked participants their and their partner's earnings in case they gave 4 Monetary Units (MU) to their partner while their partner gave 3 MU to them. Participants could respond twice to this question. Then, we recorded participants who responded correctly in their first attempt to the comprehension question. Also in this case, we did not find any differences in the results when controlling for people that responded correctly in their first attempt (Supplementary Information, section 1.1.2).

**Demographics.** Participants were also asked to give information about their gender, the highest level of education completed (i.e., elementary school, middle school, high school, some college bachelor degree, graduate school or higher) and age.

**Preregistration.** Hypotheses and design were preregistered at https://osf.io/68wds/. The data that are relevant to the analyses described herein are also publicly available[57]. Since the focus of the paper is on the variability and ubiquity of national parochialism and cooperation, we decided to report a subset of predictions presented in the preregistration. Therefore, in the current version of the manuscript we do not report results from some of the other measures we mentioned in the pre-registration. Those measures are: national identity, positive reciprocity and negative reciprocity. An additional change to the preregistered hypotheses is the inclusion of additional analyses on the role of cultural distance, historical prevalence of medieval western church and kinship norms as predictors of national parochialism and cooperation around the world. We did not originally include these as we were not aware of these indicators at the time of pre-registration (e.g., paper was not published[20]). However, considering that those indicators are hypothesized to explain global variation in national parochialism and cooperation, we included those variables in our models.

**Analytic strategy.** For the main treatment effects (partner's nationality and public vs. private), we used mixed-effects models where participants (level 2) and nations (level 3) are two random factors predicting differences in cooperation with the national ingroup compared to national outgroups. These models consider random intercepts for participants nested in nations. Also, we included partner's nationality as random slope. We decided this after comparing models through the Akaike

information criteria and the Bayesian information criteria[58]. We analyzed data with $R$ 4.0.5 (lme4 package 1.1-26) and used random intercept and slopes[59]. Meta-analyses were conducted using the package metafor (2.4-0)[60]. We ran models through two contrasts that were level-1 predictors of our models: Contrast 1 (ingroup vs. outgroup and strangers), and Contrast 2 (outgroup vs. strangers). Contrast 1 was relevant IV to test the prediction of national parochialism, while Contrast 2 allows to test whether national parochialism is driven more by a positive bias toward group members, or derogation toward outgroup members. Private vs. public choice was another level-1 predictor in the models. Individual differences variables (e.g., age, and gender) were level-2 controls. Nation was a level-3 factor.

Therefore, our model can be described by the following equation ($Y_{ijk}$ is cooperation or expected cooperation). Cooperation and Expectations range from 0 to 10:

$$\text{Level 1: } Y_{ijk} = \beta_{0jk} + \beta_{1jk}\text{CONTRAST1}_{ijk} + \beta_{2jk}\text{OBSERVABILITY}_{ijk} + \beta_{3jk}\text{CONTRAST1}_{ijk}\text{OBSERVABILITY}_{ijk} + e_{ijk}$$
$$\text{Level 2: } \beta_{0jk} = \gamma_{00k} + \gamma_{01k}\text{AGE}_{jk} + \gamma_{02k}\text{GENDER}_{jk} + f_{0jk};$$
$$\beta_{1jk} = \gamma_{10k} + \gamma_{11k}\text{AGE}_{jk} + \gamma_{12k}\text{GENDER}_{jk} + f_{1jk}$$
$$\text{Level 3: } \gamma_{00k} = \delta_{000} + g_{0k}$$
$$\gamma_{10k} = \delta_{100} + g_{1k}$$

(1)

Regarding the cross-national analyses, we ran mixed-effects models where the variable "nations" was a random factor. Those models include nation-level indicators as level 2 factors.

**Reporting Summary**. Further information on research design is available in the Nature Research Reporting Summary linked to this article.

## Data availability

All data that are relevant to the analyses described herein are available at OSF: https://osf.io/68wds/[57]. Data on nation-level indicators are available in the following publicly available sources, bilateral cultural distances: http://culturaldistance.com/, GDP per capita, GINI, and government effectiveness: http://data.worldbank.org/indicator/, rule of law: https://freedomhouse.org/report/freedom-world. Religion, self-expression values, and confidence in institutions indicators can be retrieved from the World Value Survey (Time Series database 1981–2020): https://www.worldvaluessurvey.org/WVSDocumentationWVL.jsp. The historical prevalence of infectious diseases indicator was retrieved from the Appendix of the paper from Murray and Schaller (2009)[33], tightness and looseness data were retrieved from Table 1 of the paper from Uz (2015)[61]. Intellectual autonomy and hierarchical values can be downloaded here: https://doi.org/10.13140/RG.2.1.3313.3040[62]. Data on exposure to Western Church are available here: https://doi.org/10.5061/dryad.2rbnzs7hs[63]. The Hofstede dimensions (individualism, uncertainty avoidance, power distance, long-term orientation, masculinity, and indulgence) were retrieved here: www.hofstede-insights.com/ product/compare-countries/, while country-level data on relational mobility were retrieved from: http://relationalmobility.org/. Additional data on self-expression values were retrieved from the European Value Survey (EVS Trend File 1981–2017) and are available here: https://doi.org/10.4232/1.13736[64]. The indicator "Years of life lost to communicable diseases" was retrieved from the World Health Organization (YLL estimates, 2000–2019, By Country): https://www.who.int/data/gho/data/themes/mortality-and-global-health-estimates/global-health-estimates-leading-causes-of-dalys. The Human Development Index was retrieved from: http://hdr.undp.org/en/content/human-development-index-hdi. Economic preferences indicators (i.e., trust, risk, patience, positive and negative reciprocity, altruism) were retrieved from: https://www.briq-institute.org/global-preferences/home. Source data are provided with this paper.

## Code availability

The code to reproduce the analyses described herein is available in the Supplementary Information ("R-codes").

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

## Acknowledgements
The authors gratefully acknowledge financial support under the Institutional Strategy of the University of Cologne within the German Excellence Initiative (Hans Kelsen-Prize), from Deutsche Forschungsgemeinschaft (DFG, German Research Foundation) under Germany´s Excellence Strategy – EXC 2126/1– 390838866, the European Research Council Starting Grant 635356, the Asian Office of Aerospace Research and Development Grant FA 2386-15-1-0003, the Japan Society for the Promotion of Science Grant 15H05730, and the Max Planck Institute for Research on Collective Goods. The authors thank Caroline Graf for support in retrieving part of the cross-national indicators data. The authors thank Ernst Fehr, Simon Gächter, Robert Böhm, Hannes Rush, Andy Delton, Michel Maréchal, and Michelle Gelfand for valuable comments to an earlier version of this paper.

## Author contributions
A.R., M.S., J.H.L., T.Y., and D.B. designed research; A.R., M.S., J.H.L., and D.B. performed research; A.R., J.H.L., and D.B. had access to the data; A.R. analyzed the data; and A.R., M.S., J.H.L., and D.B. wrote the paper.

## Funding

## Competing interests
The authors declare no competing interests.
