## [Peer Review File · Nature Communications]

Reviewer #1 (Remarks to the Author):

Report on NCOMMS-20-21895 Romero et al

This paper conducts experiments in 42 countries with 18'000+ participants to test ingroup cooperation (opponent is from own nation) vs outgroup cooperation (opponent is from other country) in a prisoner's dilemma. In all countries ingroup cooperation is stronger than outgroup cooperation, which is evidence for the ubiquity of parochialism. The level of cooperation varies across societies and is correlated to society-level characteristics such as ancestral distance, prevalence of infectious diseases, exposure to western churches, individualism, etc.

This is an interesting and well written paper based on an impressive data set from a culturally diverse set of countries. To my knowledge, this is the largest data set of its kind. The research question is important, and the results are interesting (although, maybe, a bit underexplored – see below). Given the meta-analysis by Balliet et al (Psych Bull 2014) I was not surprised to see parochialism in WEIRD countries, but I was surprised that parochialism (ingroup vs outgroup cooperation) is so similar across very diverse and non-WEIRD societies. The paper has the potential to become influential.

Main comments:

1. I find the figures (world maps) not very informative. I think it would be better to show a forest plot (as currently is only shown in the SI), similar to e.g., Cohn et al, Science 2019, Fig. 1, which integrates the three relevant average cooperation rates in ingroup and outgroup (identified and stranger). This would achieve two things: We would see the variation in cooperation across societies and we would see that the difference between ingroup and outgroup/stranger cooperation is similar in all countries; it would also show the averages in outgroup & stranger cooperation. You could sort this figure according to overall size of cooperation levels and combine it with the analysis of Fig. S1. So most relevant results could be in one figure.
2. Because individuals made ingroup and outgroup decisions (within subjects), the difference between ingroup and outgroup cooperation has a distribution too: some individuals likely did not distinguish, some did. The *average* parochialisms are somewhat similar (Fig. S1), but how about the variation of *individual* parochialism as displayed within countries?
3. The cultural groupings are not clear from the main text and figures.
4. I think cultural distance could be explored more. See, for example, the new dataset of cultural similarity by Muthukrishna et al, Psych Science 2020, which, in my view, is by far the best available data on cultural differences/similarities (see also <http://culturaldistance.muth.io/>). It allows calculating bilateral cultural distances, which could be linked to parochialism.
5. The results on observability (private vs public decisions) are part of the main experimental manipulation and pre-registered hypotheses, so they deserve to be reported in more detail in the paper. Why not lift Fig. S2 from the SI to the main text?
6. It is indeed surprising that Venezuela and Sweden are similar in terms of parochialism (lines 161-163). More generally, the overall fairly low variability of parochialism is surprising. Maybe this has to do with the fact that all subject pools here are from large scale and relatively developed societies, at least compared to small-scale societies. It would be useful to (at least discursively) benchmark the results here with parochialism from small-scale societies, where there seems to be evidence for ingroup favoritism related to some institutional variables. See Hruschka et al, Human Nature 2014; doi: 10.1007/s12110-014-9217-0 (not cited); and Lang et al (ref. #19). Both of them use experimental games (although not PDs).

Some minor comments:

7. 42 countries is impressive. How did you select them? How representative of the world are these countries?
8. Line 70: "certain social institutions" is very vague.
9. Lines 78-80: Isn't this a repetition of line 75?
- 10, Line 239: Give a reference to the Harris Panel. I had never heard of it before.

Reviewer #2 (Remarks to the Author):

This is a fascinating paper exploring how national parochialism and impersonal cooperation vary around the globe. The authors report a large study involving 42 countries and over 18,000 subjects, who played an hypothetical prisoner's dilemma with a stranger or with someone in the same vs. a different country. The main results are that cooperation varies substantially across the globe, while national parochialism does not: it is pervasive across countries. The authors report also a number of analyses exploring the effect of ancestral distance, quality of institutions, exposure to pathogens, etc. The authors also explore the effect of observability in decisions, finding that observability affects impersonal cooperation but not national parochialism.

I honestly loved this paper. I think it makes an excellent contribution. I am aware that the bar for publishing in Nature Communications is very high, but I think that this paper is above that bar: the results are based on a very large sample, the analysis is correct and very detailed, the topic is fascinating and timely.

I only have a few minor comments that I hope can help improve the paper:

- Regarding how important is cooperation for COVID-19 pandemic response, the "perspective article" published by Van Bavel et al. in Nature Human Behaviour may be a useful reference.
- Line 130. Not clear what statistical test you're using. A simple regression? Perhaps meta-analysis would be better, so to have also information about heterogeneity across countries?
- Lines 134-136. Can you tell more about the 10 cultural groups?
- Lines 179: this is strange, because distance between societies is a symmetric variable, while difference in cooperation is not. In other words, what does the sign of b represent? (Perhaps I'm missing something, but I found this part quite unclear).
- I agree that this data suggests that national parochialism is probably driven by something that is culture independent. I was expecting some discussion about the "need to belong" and "terror management theory". The authors might considering adding a paragraph along these lines.

Signed: Valerio Capraro

References

Van Bavel, J. J., et al (2020). Using social and behavioural science to support COVID-19 pandemic response. Nature Human Behaviour.

REVIEWER COMMENTS

Reviewer #1 (Remarks to the Author):

Report on NCOMMS-20-21895 Romero et al

This paper conducts experiments in 42 countries with 18'000+ participants to test ingroup cooperation (opponent is from own nation) vs outgroup cooperation (opponent is from other country) in a prisoner's dilemma. In all countries ingroup cooperation is stronger than outgroup cooperation, which is evidence for the ubiquity of parochialism. The level of cooperation varies across societies and is correlated to society-level characteristics such as ancestral distance, prevalence of infectious diseases, exposure to western churches, individualism, etc.

This is an interesting and well written paper based on an impressive data set from a culturally diverse set of countries. To my knowledge, this is the largest data set of its kind. The research question is important, and the results are interesting (although, maybe, a bit underexplored – see below). Given the meta-analysis by Balliet et al (Psych Bull 2014) I was not surprised to see parochialism in WEIRD countries, but I was surprised that parochialism (ingroup vs outgroup cooperation) is so similar across very diverse and non-WEIRD societies. The paper has the potential to become influential.

Thanks for the constructive comments about our project. We have carefully followed your suggestions to strengthen our manuscript. Find our responses below.

Main comments:

1. I find the figures (world maps) not very informative. I think it would be better to show a forest plot (as currently is only shown in the SI), similar to e.g., Cohn et al, Science 2019, Fig. 1, which integrates the three relevant average cooperation rates in ingroup and outgroup (identified and stranger). This would achieve two things: We would see the variation in cooperation across societies and we would see that the difference between ingroup and outgroup/stranger cooperation is similar in all countries; it would also show the averages in outgroup & stranger cooperation. You could sort this figure according to overall size of cooperation levels and combine it with the analysis of Fig. S1. So most relevant results could be in one figure.

Thanks for your suggestion. In the revised version, we now include a new Figure (Figure 1) which includes the average cooperation rates with ingroup and outgroup members/strangers. Moreover, as you suggested, we sorted nations based on the average cooperation levels and integrated the plot with analyses of Figure S1 including a report of the Cohen's d estimates of the national parochialism effect in each country (see p 23).

2. Because individuals made ingroup and outgroup decisions (within subjects), the difference

*between ingroup and outgroup cooperation has a distribution too: some individuals likely did not distinguish, some did. The *average* parochialisms are somewhat similar (Fig. S1), but how about the variation of *individual* parochialism as displayed within countries?*

In the revised version of the manuscript, we now include information about individual variation in parochialism. We indeed observe higher variation within countries ($SD_{\text{between}} = 0.13$, $SD_{\text{within}} = 1.25$). We now elaborate more about individual variation in national parochialism in the results section (p. 8). We also advance a few potential individual difference variables which may explain these differences. In fact, we found that men, compared to women, cooperate more with ingroup members than outgroup members and strangers ($b = 0.03$, $p = .04$). We also found that education moderates the effect of national parochialism, such that people with higher education show less national parochialism, compared to people with lower education ($b = -0.02$, $p = .02$).

3. The cultural groupings are not clear from the main text and figures.

This is a good point. We formed cultural groupings based on previous seminal work by Inglehart and Baker (2000) which divide countries across two dimensions: “traditional vs secular values” and “survival vs self-expression values”. Moreover, similar to previous research (Gächter, Herrmann, & Thöni, 2008), for Arabic speaking and Southern European countries we referred to Hofstede (2001). We now clarify how cultural groupings were formed and list each category and the correspondent nations on page 11 and 12 of the Supplementary information. We also refer the reader to that in the manuscript on page 7.

4. I think cultural distance could be explored more. See, for example, the new dataset of cultural similarity by Muthukrishna et al, Psych Science 2020, which, in my view, is by far the best available data on cultural differences/similarities (see also <http://culturaldistance.muth.io/>). It allows calculating bilateral cultural distances, which could be linked to parochialism.

Thanks for this suggestion. We collected data from this additional dataset and, in line with what we find with other indicators (e.g., ancestral distance), we found that cultural distance does not predict national parochialism ($b = -0.22$, $p = .70$). By contrast, cultural distance between pair countries predicts bilateral distances in cooperation ($b = 0.59$, $p < .001$). We now report the results of these analyses in the manuscript (p. 7, p. 9).

5. The results on observability (private vs public decisions) are part of the main experimental manipulation and pre-registered hypotheses, so they deserve to be reported in more detail in the paper. Why not lift Fig. S2 from the SI to the main text?

Following your suggestion, we now include the results of Figure S2 in the manuscript (p. 24) and elaborate more on the observability findings in the results section (pp. 7-8).

6. *It is indeed surprising that Venezuela and Sweden are similar in terms of parochialism (lines 161-163). More generally, the overall fairly low variability of parochialism is surprising. Maybe this has to do with the fact that all subject pools here are from large scale and relatively developed societies, at least compared to small-scale societies. It would be useful to (at least discursively) benchmark the results here with parochialism from small-scale societies, where there seems to be evidence for ingroup favoritism related to some institutional variables. See Hruschka et al, Human Nature 2014; doi: 10.1007/s12110-014-9217-0 (not cited); and Lang et al (ref. #19). Both of them use experimental games (although not PDs).*

Thanks for raising this interesting point. Indeed, it is true that the low variability can be a feature of relatively large-scale societies, compared to small-scale societies. That said, differences in the observed variation could be dependent on the specific interdependent situation where individuals make decisions (cooperation vs generosity). Indeed, previous work has found that the degree of ingroup favoritism is stronger in social dilemma situations (like we use here), compared to dictator games and trust games (Balliet et al., 2014). We think this is an open question for future research. We followed your suggestion and now elaborate about these potential differences in the conclusions section (pp. 10-11).

Some minor comments:

7. *42 countries is impressive. How did you select them? How representative of the world are these countries?*

Within constraints of the available countries of the panel, we aimed at getting the most diversified set of countries in terms of quality of institutions (e.g., rule of law). Altogether, these 42 nations capture 73% of the world population. We now included a sentence with this information at the end of page 4.

8. *Line 70: “certain social institutions” is very vague.*

We changed “certain social institutions” to “cultural differences in social institutions like rule of law and modern world religions”.

9. *Lines 78-80: Isn't this a repetition of line 75?*

Good point, we removed the repetition at lines 78-80.

10, *Line 239: Give a reference to the Harris Panel. I had never heard of it before.*

We now include a reference to the Harris Panel (p. 13).

Reviewer #2 (Remarks to the Author):

This is a fascinating paper exploring how national parochialism and impersonal cooperation vary around the globe. The authors report a large study involving 42 countries and over 18,000 subjects, who played an hypothetical prisoner's dilemma with a stranger or with someone in the same vs. a different country. The main results are that cooperation varies substantially across the globe, while national parochialism does not: it is pervasive across countries. The authors report also a number of analyses exploring the effect of ancestral distance, quality of institutions, exposure to pathogens, etc. The authors also explore the effect of observability in decisions, finding that observability affects impersonal cooperation but not national parochialism.

I honestly loved this paper. I think it makes an excellent contribution. I am aware that the bar for publishing in Nature Communications is very high, but I think that this paper is above that bar: the results are based on a very large sample, the analysis is correct and very detailed, the topic is fascinating and timely.

I only have a few minor comments that I hope can help improve the paper:

- Regarding how important is cooperation for COVID-19 pandemic response, the "perspective article" published by Van Bavel et al. in Nature Human Behaviour may be a useful reference.

Thanks for this suggestion, we included the reference at page 3.

- Line 130. Not clear what statistical test you're using. A simple regression? Perhaps meta-analysis would be better, so to have also information about heterogeneity across countries?

Thanks for this comment, we used a mixed-effects regression (with participants and countries as random factors), which take into account heterogeneity across countries (and among participants nested in countries). We now include this information in a revised sentence (page 7).

- Lines 134-136. Can you tell more about the 10 cultural groups?

This comment was also made by Reviewer 1. We formed cultural groupings based on previous seminal work by Inglehart and Baker (2000) which divide countries across two dimensions: "traditional vs secular values" and "survival vs self-expression values". Moreover, similar to previous research (Gächter, Herrmann, & Thöni, 2010), Culture and cooperation), for Arabic speaking and Southern European countries we referred to Hofstede (2001). We now clarify how cultural groupings were formed and list each category and the correspondent nations at pages 11-12 of the Supplementary information. We also refer the reader to that in the manuscript at page 7.

- Lines 179: *this is strange, because distance between societies is a symmetric variable, while difference in cooperation is not. In other words, what does the sign of b represent? (Perhaps I'm missing something, but I found this part quite unclear).*

Thanks for drawing our attention to this aspect of the manuscript. We computed absolute differences in cooperation levels and then related them to absolute ancestral distances. In the revised SI (page 18), we have clarified how we computed each indicator and which analysis we run. This should clarify the interpretation of the coefficient for the reader.

- *I agree that this data suggests that national parochialism is probably driven by something that is culture independent. I was expecting some discussion about the “need to belong” and “terror management theory”. The authors might considering adding a paragraph along these lines.*

Thanks for this suggestion. It is true that in the previous version of the manuscript we did not speculate much about potential drivers of national parochialism. As we did not test any specific hypotheses related to these theories, and since we focus more on potential functional explanations of ingroup favoritism, we would prefer to avoid making a reference to these theories. In the revised version of the conclusions, we added a sentence about how our findings can support theories which hypothesize that ingroup favoritism is pervasive in humans (p. 10).

Signed: Valerio Capraro

References

Van Bavel, J. J., et al (2020). Using social and behavioural science to support COVID-19 pandemic response. Nature Human Behaviour.

Reviewer #1 (Remarks to the Author):

Report on Romano et al

This is a revised version that incorporates the changes I suggested. I am largely satisfied with the changes made and think they have improved the paper. I have a few comments on the new version that the authors should consider:

- 1) P. 5, line 92: Stranger condition: According to line 290 and the instructions in the Supplementary Information, nationality was not mentioned at all. Thus, the Stranger condition leaves it open whether the opponent is ingroup or outgroup. This should be made clear here (and not just in the instructions and line 290, which is also slightly ambiguous) because another possibility would have been to tell participants that their opponent is of a different, though unknown, nationality.
- 2) Figure 1 is very interesting and more informative than the previous version. However, Outgroup and Strangers are combined here. Is that justified because they are virtually identical everywhere? Shouldn't there be a test for this rather than collapsing these two conditions throughout the whole analysis?
- 3) At least in principle, given this design, there are two types of "parochialisms" here: an identified (towards a known nationality) and an unidentified one (towards a stranger who could be own or other nationality). Why not show these results as well (e.g., by a green open dot) and to calculate Cohen's d for both types of parochialisms?
- 4) Another issue with Fig. 1 is that it is unclear whether it contains the data from observable and non-observable (public vs. private) treatments. Thus, the caption should make clear what exactly is displayed here. The sorting criteria should also be explained in the caption.
- 5) Line 141: The fact that cultural distance does not seem to explain the extent of parochialism is a really remarkable result. If I understand this correctly, this means, e.g., that an American cooperates about the same with a culturally close Canadian than with a culturally distant Kenyan or Venezuelan. I think this remarkable result of culturally non-discriminatory parochialism deserves to be illustrated and documented a bit more, at least in the supplementary materials, where it currently is totally absent (see last comment below).
- 6) The supporting statistical analyses in the Supplementary Information should provide more details on the exact statistical approaches used. Participants made twelve decisions, how is this controlled for? In principle, the data set is massive: $18,411 \times 12 = 220,932$ data points. But when I look at the tables, they seem to have far fewer observations. Why is that? I suggest being a bit more specific about this and to include the number of observations that enter a regression.
- 7) A general observation is that none of the Supplementary Tables are mentioned in the main text.
- 8) Ancestral distance analysis in the Supplementary Information: Why not show the regression model/output here?
- 9) The cultural distance analysis mentioned in lines 140/141 is missing in the Supplementary Information. This should be explained in at least as much detail as the ancestral distance analysis.

Reviewer #2 (Remarks to the Author):

Thanks for addressing all my comments. I am looking forward to seeing this paper published.

Reviewer #1 (Remarks to the Author):

Report on Romano et al

This is a revised version that incorporates the changes I suggested. I am largely satisfied with the changes made and think they have improved the paper. I have a few comments on the new version that the authors should consider:

We are glad to read that you think the previous round of revision improved the paper and thanks for the additional comments which helped us clarify some aspects of the design, results and statistical approach used in the study.

1) P. 5, line 92: Stranger condition: According to line 290 and the instructions in the Supplementary Information, nationality was not mentioned at all. Thus, the Stranger condition leaves it open whether the opponent is ingroup or outgroup. This should be made clear here (and not just in the instructions and line 290, which is also slightly ambiguous) because another possibility would have been to tell participants that their opponent is of a different, though unknown, nationality.

Thanks for pointing this out. We now clarify this in the manuscript in the section where we explain the experimental design (page 5). We now define the stranger condition as “person without any information provided about their nationality”.

2) Figure 1 is very interesting and more informative than the previous version. However, Outgroup and Strangers are combined here. Is that justified because they are virtually identical everywhere? Shouldn't there be a test for this rather than collapsing these two conditions throughout the whole analysis?

This is a good point and in the previous version of the manuscript we did not clarify why we made this decision. We combined the Outgroup and Stranger treatments for theoretical reasons (as also mentioned in our preregistration: <https://osf.io/gnxv2/>). Based on previous research (Balliet, Wu, & De Dreu, 2014; *Psychological Bulletin*), we implemented these two different treatments to disentangle two motives behind national parochialism. On one hand the contrast of Ingroup vs Outgroup and Strangers is a direct test of Ingroup favoritism. On the other hand, a comparison between Outgroup and Strangers can shed light on whether national parochialism is driven by outgroup derogation. In line with previous research, we found evidence for parochialism being driven by ingroup favoritism and not outgroup derogation (see Supplementary information, section 1.8, page 14). We now mention the two different underlying motivations in the revised manuscript (see the results section, page 7).

Additionally, we found that people cooperate more with outgroup members than strangers, and the effect size of this difference ($d = 0.14$) falls within the *Prediction Interval* (i.e., the range of values of effect sizes predicted for future studies) of the

outgroup-versus-stranger comparison estimated by a recent meta-analysis (Balliet et al., 2014). Importantly, a data-driven approach would suggest to combine the outgroup and stranger condition. In fact, when we run a principal component analysis of the 12 decisions, the outcome of these analyses suggest one of the components which explained most variance was the one where the outgroup and strangers treatments loaded on the same dimension (vs Ingroup), providing additional independent support for combining outgroup and strangers when studying national parochialism in cooperation.

Therefore for both theoretical and empirical reasons, we would prefer to keep Figure 1 in the manuscript as it is.

That said, we now include two Cleveland plots including the effect sizes in each society for each of the comparisons described above (ingroup vs outgroup, and ingroup vs stranger) in the supplementary information (SI, pages 16 and 17).

3) At least in principle, given this design, there are two types of “parochialisms” here: an identified (towards a known nationality) and an unidentified one (towards a stranger who could be own or other nationality). Why not show these results as well (e.g., by a green open dot) and to calculate Cohen’s d for both types of parochialisms?

For the reasons expressed above, we would prefer to incorporate outgroup and strangers when reporting the main results of national parochialism in the manuscript. Importantly, this would also be more coherent with our pre-registered analyses: <https://osf.io/gnxv2/>. That said, we agree that reporting these results can be interesting, and therefore we now include two Cleveland plots including the effect-sizes in each country for each of the two outgroup treatments (ingroup vs outgroup, and ingroup vs stranger) in the supplementary information (pages 16 and 17).

4) Another issue with Fig. 1 is that it is unclear whether it contains the data from observable and non-observable (public vs. private) treatments. Thus, the caption should make clear what exactly is displayed here. The sorting criteria should also be explained in the caption.

Thanks for raising this. Figure 1 includes both the public and private treatments and countries are sorted based on their average cooperation levels. In the revised version of the manuscript we clarify these aspects in the caption of Figure 1.

5) Line 141: The fact that cultural distance does not seem to explain the extent of parochialism is a really remarkable result. If I understand this correctly, this means, e.g., that an American cooperates about the same with a culturally close Canadian than with a culturally distant Kenyan or Venezuelan. I think this remarkable result of culturally non-discriminatory parochialism deserves to be illustrated and documented a bit more, at least in the supplementary materials, where it currently is totally absent (see last comment below).

The cultural distance result suggests something slightly different than what you wrote. Our results show that bilateral cultural distances do not predict differences in how people discriminate in favor of their nationalities, compared to other nationalities. This means that a Canadian shows the same extent of favoritism toward the own nation as a distant Kenyan or Venezuelan. That said, we agreed that this result deserves more space. In the revised version of the manuscript, we now clarify the interpretation of these results at page 7, and provide more details on the cultural distance analysis in the Supplementary Information (pages 20-21).

6) The supporting statistical analyses in the Supplementary Information should provide more details on the exact statistical approaches used. Participants made twelve decisions, how is this controlled for? In principle, the data set is massive: $18,411 \times 12 = 220,932$ data points. But when I look at the tables, they seem to have far fewer observations. Why is that? I suggest being a bit more specific about this and to include the number of observations that enter a regression.

We used mixed-model regressions (multi-level models) with Participants and Countries as random intercepts, and the effect of national parochialism as random slope. Hence, our statistical analyses take into account dependencies within individuals (i.e., 12 decisions) and within countries (i.e., 42 countries). Following your suggestions, we now include more information on the number of observations in each statistical model reported in the Supplementary information (See revised SI). Details on the statistical approach can be found in the analytic strategy in the methods section (page 16-17), and in the supplementary information (e.g., R-code, SI pages 22 to 24).

7) A general observation is that none of the Supplementary Tables are mentioned in the main text.

Thanks for spotting this, we now include a sentence with a reference to all the supplementary tables in the results section of the paper (page 7).

8) Ancestral distance analysis in the Supplementary Information: Why not show the regression model/output here?

Done, thanks (SI pages 20-21).

9) The cultural distance analysis mentioned in lines 140/141 is missing in the Supplementary Information. This should be explained in at least as much detail as the ancestral distance analysis.

Done, thanks. We have revised the ancestral distance section in the supplementary information to include a description of both the ancestral and the cultural distance (pages 20-21).

Reviewer #2 (Remarks to the Author):

Thanks for addressing all my comments. I am looking forward to seeing this paper published.

Reviewer #3

Here are some thoughts on the Romano et al. paper, with special reference to the 'ancestral distance' measure.

First, that measure. The measure itself is of unknown value. It has what would be termed 'face validity' in the sense that it seems reasonable to assume that greater ancestral distance will, very broadly, probably be associated with greater amount of time since two countries shared a common ancestor. It is not what the authors say it is (see SI, lines 320-321): "ancestral distance is a measure of the length of the time elapsed since a common ancestor of respective groups broke apart". It could be thought of as a very crude indicator of this, but that is all. It will be full of error, such that two countries with differing ancestral distances might in fact have separated in the reverse order. What they really want to measure here is something like national-kinship: are there other countries one feels a greater "kinship" to independently of this distance measure.

So, credit to them for trying, but one shouldn't assume that the measure is anything other than a very very crude indicator. It combines genetic and linguistic evidence equally (by standardising them first): it's not obvious that this is the right thing to do.

My point is not to fault the authors – they have tried – but this shouldn't be treated as anything more than lip service to the question. On the other hand, it is not really a critical part of their paper.

I have more serious reservations about the methodology and findings more generally. The study risks being heavily contaminated with what are known as 'demand characteristics' or all of the cues that tip off subjects as to what the experimenters are looking for. Subjects also want to look good. The authors' methodology explicitly and openly varies whether the person you are interacting with is from your country or not and this won't be lost on the subjects. Their public vs private condition shows that there is more cooperation when the interactions are public – this is not surprising as subjects are motivated to look good.

Even without worries about demand characteristics and motivations to look good, the same vs other country manipulation is confounded with "uncertainty". Subjects know what the norms of cooperation are within their own countries but will be less certain of these norms when interacting with subjects from other countries. So, think of this experiment as contrasting two investment decisions: one where there is far more uncertainty than the other. Independently of whether someone is from your country, uncertainty will make you more risk averse, that is, you will give less.

Had the authors found big effects from their manipulations, one might be able to look the other way. But the effect of same vs other country or same vs other+stranger is only around 0.3 MUs. In terms of their example with US dollars, this amounts to about a 48 cent difference or, using their figures about 45 seconds of work (incidentally, this is based on an assumption that the average pay in the US for an hour's work is approximately \$28 – I would have thought it lower). This is not a large effect by any means and could easily be explained by demand characteristics and the desire to look good. Their stranger condition acts like the different country condition because participants know that they are playing with people from all over the world. Your best guess then if a person is a stranger is that they are not from your country. So, their conclusion about favouring own nation vs derogating others is inconclusive.

Social science investigators began to turn away from the sort of methodology the current authors use some years ago because of the problems I list above. The current authors, as far as I can tell, didn't even debrief their subjects to find out what they thought the purpose of the study was. They should have done.

REVIEWER COMMENTS

Reviewer #3

Here are some thoughts on the Romano et al. paper, with special reference to the ‘ancestral distance’ measure.

First, that measure. The measure itself is of unknown value. It has what would be termed ‘face validity’ in the sense that it seems reasonable to assume that greater ancestral distance will, very broadly, probably be associated with greater amount of time since two countries shared a common ancestor. It is not what the authors say it is (see SI, lines 320-321): “ancestral distance is a measure of the length of the time elapsed since a common ancestor of respective groups broke apart”. It could be thought of as a very crude indicator of this, but that is all. It will be full of error, such that two countries with differing ancestral distances might in fact have separated in the reverse order. What they really want to measure here is something like national-kinship: are there other countries one feels a greater “kinship” to independently of this distance measure.

So, credit to them for trying, but one shouldn’t assume that the measure is anything other than a very very crude indicator. It combines genetic and linguistic evidence equally (by standardising them first): it’s not obvious that this is the right thing to do.

My point is not to fault the authors – they have tried – but this shouldn’t be treated as anything more than lip service to the question. On the other hand, it is not really a critical part of their paper.

Thanks for this comment. We agree that the ancestral distance measure did not represent a critical part of the paper. This measure simply provided additional evidence for the claim that there is higher between-country variation in cooperation, compared to between-country variation in national parochialism. That said, we appreciate and understand the points raised by the Reviewer and agree that our measure may be a “crude indicator”. Therefore, we decided to remove the ancestral distance measure entirely from the analyses and paper.

I have more serious reservations about the methodology and findings more generally. The study risks being heavily contaminated with what are known as ‘demand characteristics’ or all of the cues that tip off subjects as to what the experimenters are looking for. Subjects also want to look good. The authors’ methodology explicitly and openly varies whether the person you are interacting with is from your country or not and this won’t be lost on the subjects. Their public vs private condition shows that there is more cooperation when the interactions are public – this is not surprising as subjects are motivated to look good.

We understand the concerns raised by the Reviewer, and we considered these issues when designing the study. We believe that our findings cannot be explained by the ‘demand characteristics’ that the Reviewer is referring to for the following reasons:

- 1) Parochial cooperation does not differ between experiments that manipulate partner group membership by using either a between-subjects or a within-subject manipulation. A recent meta-analysis (Balliet, Wu, & De Dreu, 2014, *Psych. Bull.*) investigated potential demand characteristics in studies on parochial cooperation by testing whether the effect sizes for parochial cooperation (i.e., differences in cooperation with an ingroup member versus an outgroup member) would differ depending on whether the experiment manipulated partner group membership using either a between-subjects design (70 studies) or within-subjects design (55 studies). Results from this meta-analysis show that there is no difference in the degree of parochial cooperation across these experimental designs. This evidence suggests that when participants interact with several partners (including both ingroup and outgroup partners) they do not behave differently, such as by showing greater discrimination, for example, compared to experiments where participants are only assigned to interact with an ingroup member or an outgroup member/stranger.
- 2) Moreover, if the participants’ behaviors in our study was primarily driven by a motivation “to look good”, we would not expect to observe national discrimination (e.g. greater cooperation toward ingroup members compared to outgroup and strangers), a behavior that is socially *undesirable* – at least in some countries. Importantly, people showed the same degree of national parochialism in both public and private situations, suggesting that a motivation for impression management does not affect how people behaved towards a national ingroup member, national outgroup member, or strangers in our study. Moreover, if people were merely driven by demand characteristics, we would not expect that individual differences in national identification would be associated with national parochialism. However, supporting the idea that our results are driven by a motivation for ingroup favoritism and not demand characteristics, national identification ($b = 0.10, p < .001$) was positively associated with national parochialism around the globe. We now elaborate about these findings in the Supplementary Information (SI page 15).
- 3) We also have reason to believe that observability (public vs private choice) had a positive effect on cooperation due to a general reputational concern (as predicted by an indirect reciprocity perspective, e.g. Milinski et al. 2006, *PNAS*) that extends beyond a reputational concern about how they are perceived by the experimenter (i.e., experimental demand). In fact, we followed past research that used the same approach to test theory about general reputational concerns (Ariely, Bracha, & Meier, 2009, *AER*; Milinski et al. 2006, *PNAS*). Notably, we agree with the statement made by the reviewer that the observability effect is not a surprise, and that it replicates previous experimental findings observed in both US and Japan. Also, previous studies have found that observability cues are equally impactful on behavior while using either

between-subjects or within-subjects designs, which also reduces concerns that experimenter demand characteristics can explain these results (Ariely, Bracha, & Meier, 2009, *AER*; or Romano, Wu, & Balliet, 2017, *JESP*). Also, it is important to note that in our study the main purpose of including an observability treatment was to test the hypothesis driven by an indirect reciprocity perspective (as also stated in the introduction on page 3, and in our pre-registration) that national parochialism would be stronger when participant choices' were public, compared to when their choices were private (an hypothesis that we do not find support for, see results pages 7-8).

- 4) Additionally, we also decided to include incentives in three countries to provide a further test to the claim that behavior could be impacted by experimental demand. In fact, in these situations, past prominent research theorized that the use of incentives shall remove any concerns about demand characteristics (e.g., Smith, 1976, *AER*; Smith et al, 2007, *Economic Inquiry*). Therefore, if demand characteristics drive behavior in our study, we should observe a difference in behavior between the incentivized vs hypothetical treatments. However, we found the same behavioral pattern for national parochialism and observability in both incentive and hypothetical treatments, which provides additional support for the idea that people's behavior was not driven by guessing the experimental design and a motivation to look good to the experimenter (page 6). We now mention this in the Supplementary information (page 7). Finally, we would like to note that a very recent paper (De Quidt, Haushofer, & Roth, 2018; *AER*) carefully investigated the issue of experimenter demand effects, by also investigating beliefs about the experimenter's hypotheses. Again, the authors found low and not different demand effects across within-subjects and between-subjects designs.

In summary, we believe that our experimental design and data robustly provide support for the existence of ingroup favoritism across societies and make us confident that our results (for both observability and national parochialism) are not driven by demand characteristics. This is supported by several sources of information, including (1) a past meta-analysis; (2) the observed relation in our study between national identification and national parochialism; (3) that the manipulation of incentives in our study did not moderate how partner group membership and observability affected behavior. We now elaborate about these issues in the supplementary information (see SI page 7 for a discussion about the incentive results with additional reference to previous findings on between vs within-subjects designs, and SI page 15 for a report of the analysis on national parochialism and national identification). By contrast, we are confident that a more plausible explanation is offered by the mediator we consider in our main analysis: expectations of partner cooperation (see results page 8). Specifically, we found that expectations partially mediated the relation between partner group membership and cooperation around the globe.

Even without worries about demand characteristics and motivations to look good, the same vs other country manipulation is confounded with "uncertainty". Subjects know what the norms

of cooperation are within their own countries but will be less certain of these norms when interacting with subjects from other countries. So, think of this experiment as contrasting two investment decisions: one where there is far more uncertainty than the other. Independently of whether someone is from your country, uncertainty will make you more risk averse, that is, you will give less.

This is a fascinating point and potential explanation for why we observe greater cooperation with ingroup members, compared to outgroup members and strangers. The reviewer suggests that people are more uncertain about outgroup norms of cooperation and this explains why people make different decisions to cooperate across the two treatments. However, several pieces of evidence lead us to think that a more plausible explanation for our findings is that people hold positive expectations toward ingroup members. We found evidence for a similar amount of national parochialism across countries that are actually characterized by very different norms of cooperation and trust. If “uncertainty” about the norms between vs. within countries drives national parochialism, we would have expected much stronger differences in the effect of national parochialism – and this would be predicted by cross-national differences in cooperative norms (e.g., norms of civic cooperation and trust). Yet, we do not find such differences (see results page 8). Moreover, previous research has found that people hold different norms across countries about cooperating with the fellow citizens, or with ingroup members (e.g., Hruschka & Henrich, 2013, *Frontiers in Human Neuroscience*). However, we observe that people cooperate more with ingroup, compared to outgroup members and strangers, to a similar extent across all of these countries which differ in norms of cooperation.

Finally, if it was true that there is more uncertainty in interactions with outgroup members and strangers compared to ingroup members, we would expect to observe higher variability in expectations and cooperation decisions with outgroup members and strangers, compared to ingroup members. We checked the standard deviation within each treatment in our dataset. An uncertainty perspective would imply greater variation in expected partner cooperation and the participants’ own cooperation for interactions with outgroup members and strangers, compared to ingroup members. We found that the standard deviations of expectations ($sd = 2.60$) and cooperation ($sd = 2.64$) in decisions with outgroup members and strangers were actually slightly lower than the standard deviations for expectations ($sd = 2.66$) and cooperation ($sd = 2.72$) in decisions with ingroup members. This evidence does not support the idea that uncertainty causes the differences in cooperative behavior with national ingroup members compared to the other treatments. Therefore, altogether, based on what we would expect from cross-cultural research on norms and our data, we strongly believe that our data are neither driven by experimental demands or uncertainty about norms when interacting with outgroup members.

Had the authors found big effects from their manipulations, one might be able to look the other way. But the effect of same vs other country or same vs other+stranger is only around 0.3 MUs. In terms of their example with US dollars, this amounts to about a 48 cent

difference or, using their figures about 45 seconds of work (incidentally, this is based on an assumption that the average pay in the US for an hour's work is approximately \$28 – I would have thought it lower). This is not a large effect by any means and could easily be explained by demand characteristics and the desire to look good. Their stranger condition acts like the different country condition because participants know that they are playing with people from all over the world. Your best guess then if a person is a stranger is that they are not from your country. So, there conclusion about favouring own nation vs derogating others is inconclusive.

Although it is true that the raw differences between monetary units may appear low, the magnitude of differences are in line with previous research on national parochialism in social dilemmas. Furthermore, effect sizes observed in behavioral sciences, in general, are also of a similar size (i.e., small-to-medium magnitude, see Funder & Ozer, 2019, *Advances in Methods and Practices in Psychological Science*). In fact, our effect sizes range from small to medium size across the 42 countries. The overall effect size for national parochialism is Cohen's d of .22. As shown by a recent meta-analytic review of all studies on discrimination in social dilemmas (Balliet, Wu, & De Dreu, 2014, *Psych. Bull.*) this observed effect size falls within the prediction interval of effect sizes. This means that our study falls within the expected range of possible true values for the effect size of parochial cooperation in social dilemmas. In fact, half of the observed effect sizes are actually medium effect sizes (up to $d = 0.43$ in Venezuela) and the overall estimate was higher than our estimated effect size based on a sensitivity power analysis and a 95% power ($d = .16$).

Regarding the stranger treatment and our claim about ingroup favoritism and outgroup derogation, the Reviewer makes a good point that in the stranger condition it is more likely that people would interact with outgroup members, compared to ingroup member. That said, this is an established benchmark in the literature, and our results are in line with previous research (Balliet, Wu, & De Dreu, 2014). In fact, in the outgroup condition people are 100% sure they will not interact with ingroup members, while in the stranger condition there is at least a chance that the interacting partner is an ingroup member. That said, also in this case we took the point of the reviewer seriously and looked at our data to further validate our findings. We retrieved our survey item of national identification where participants were asked to what extent they agreed to the following statement "I identify with my nationality", from a 1 to 7 likert-scale. If it is true that cooperation with people of the own nationality is driven by ingroup favoritism, we would observe that people who scored high (low) on national identification would cooperate more (less) with ingroup members: we find support for this hypothesis ($b = 0.10$, $p < .001$; see revised SI page 15). Therefore, we are confident that our results on national parochialism are driven by ingroup favoritism.

Nonetheless, although the comparison between outgroup and stranger is a common method to disentangle between motivations for ingroup favoritism and outgroup derogation, we acknowledge that the classic prisoner's dilemma is not the ideal setting

to investigate outgroup hate/derogation. In fact, the ideal setting to study such specific motivation would be an experimental game in which people can at the same time favor the ingroup and harm the outgroup (for example, see work by De Dreu et al., 2016, *PNAS*, or Halevy, Bornstein, & Sagiv, 2008; *Psych. Science*). We added this point as a potential direction for future research (see conclusions at page 11).

Social science investigators began to turn away from the sort of methodology the current authors use some years ago because of the problems I list above. The current authors, as far as I can tell, didn't even debrief their subjects to find out what they thought the purpose of the study was. They should have done.

We understand the reviewer concerns about the potential confounds (experimental demands and uncertainty) of our approach. We carefully considered these aspects when designing our experiment. We carefully crafted our study based on previous research that found no differences across within- and between-subjects design, both for national parochialism (Balliet, Wu and De Dreu, 2014, *Psych. Bull.*) and other behaviors (e.g., d'Adda, Drouvelis & Nosenzo, 2016; *Journal of Behav. and Experim. Econ.*). Moreover, the use of within-subjects designs has the unique advantage to increase statistical power to test hypotheses.

Based on these reasons, we did not feel compelled to assess what participants thought was the purpose of the experiment. Importantly, past research found that individuals are not accurate in the interpretation of their own mental states or behavior (see classic work by Nisbett & Wilson, 1977, *Psychological Review* or Podsakoff et al., 2012, *Annual Review of Psychology*). Therefore, we believe that such questions would have been difficult to interpret, especially in our large cross-cultural sample with more than 18,000 participants and 20 different languages. That said, we believe that based on previous research and the results of our study, we have several sources of information that make us confident to rule out alternative accounts of our findings, such as demand effects and uncertainty.

Reviewer #3 (Remarks to the Author):

The authors have made the right choice in removing the cross-country ancestral distance analyses.

Regarding comments on 'demand characteristics' it is a shame that subjects -- even a subset -- were not de-briefed as I am not entirely convinced by what the authors have to say here, although it is good that they have taken this point seriously. I'd like to see some mention of this issue in the main text with readers directed to the SI if necessary for further evaluation. At present, unless I have missed something, there is no acknowledgement in the text of this as a possible alternative explanation.

The point about the average differences in MUs being low, then, becomes more prominent in the light of this possibility of 'demand characteristics' and this should be acknowledged and again discussed somewhere openly. The point of both these remarks is that the authors' research should withstand the scrutiny of the research community that is interested in these questions.

Then, even if I were to accept that the MUs are real, at the moment, the measured differences in within versus between-group cooperation are small. Should I (and every other reader) assume then that parochialism is not such a big deal? My hunch is that parochialism is in some respects a very big deal so why is the authors' methodology not detecting this? Again, some discussion of such points would greatly strengthen this contribution. This is an important point: if we accept as true everything about the methodology used here, one could almost draw exactly the opposite conclusions from this paper that the authors intend: that parochialism isn't the big problem we all thought it was. Indeed, it boils down to a measly 48 US cents.

This same comments hold for the point I made about uncertainty and risk taking. The authors should acknowledge this and discuss it.

Finally, a plea to the authors to use simpler language. In their replies to my comments, all in the space of a few lines the authors talk about "parochial cooperation", "national discrimination" and then "national parochialism"! These terms might trip off their tongues, but they are almost unreadable to anyone not in the 'ingroup' of this research area. This jargon language extends to the paper. See, for example (I could have chosen many others) lines 146-148: after a particularly heavy paragraph discussing public block, private blocks, indirect reciprocity and national parochialism, we get "This latter finding does not support the hypothesis that national parochialism is a strategy to acquire indirect benefits in a system of reputation-based indirect reciprocity that is contained within a group." What?

REVIEWER COMMENTS

The authors have made the right choice in removing the cross-country ancestral distance analyses.

Regarding comments on 'demand characteristics' it is a shame that subjects -- even a subset -- were not de-briefed as I am not entirely convinced by what the authors have to say here, although it is good that they have taken this point seriously. I'd like to see some mention of this issue in the main text with readers directed to the SI if necessary for further evaluation. At present, unless I have missed something, there is no acknowledgement in the text of this as a possible alternative explanation.

The point about the average differences in MUs being low, then, becomes more prominent in the light of this possibility of 'demand characteristics' and this should be acknowledged and again discussed somewhere openly. The point of both these remarks is that the authors' research should withstand the scrutiny of the research community that is interested in these questions.

Following your suggestion, we now include a paragraph in the conclusions (page 12) where we elaborate on potential limitations of our study, including a discussion on demand characteristics, and small effect sizes. In particular, on page 12 of the conclusions we now highlight how (a) previous literature did not find differences in parochialism across within- and between-subjects designs, suggesting that demand characteristics should not play a major role in our design (e.g., Balliet, Wu & De Dreu, 2014), and (b) that the lack of statistical difference between the hypothetical and incentivized treatments supports the hypothesis that demand-characteristics do not play a major role in our study (Smith, 1976). Finally, we draw attention to the relatively small effect sizes, and make a comparison with effect sizes observed in previous research (page 12).

Then, even if I were to accept that the MUs are real, at the moment, the measured differences in within versus between-group cooperation are small. Should I (and every other reader) assume then that parochialism is not such a big deal? My hunch is that parochialism is in some respects a very big deal so why is the authors' methodology not detecting this? Again, some discussion of such points would greatly strengthen this contribution. This is an important point: if we accept as true everything about the methodology used here, one could almost draw exactly the opposite conclusions from this paper that the authors intend: that parochialism isn't the big problem we all thought it was. Indeed, it boils down to a measly 48 US cents.

In the revised conclusions, we also include a discussion about the average differences in MU among our experimental treatments (page 12). In particular, although we acknowledge that the observed effect sizes are small to medium, we address how these effect sizes are aligned with existing research on parochialism (i.e., research that

investigates situations where people cooperate more with ingroup members, compared to outgroup members and strangers, see Balliet, Wu, & De Dreu, 2014, *Psychological Bulletin*), and refer to literature that elaborates on why small effect sizes can have meaningful, accumulative, and practical importance, because even relatively minor degrees of parochialism can have large consequences (e.g., Funder & Ozer, 2019, *AMPPS*; Abelson, 1985, *Psychological Bulletin*).

This same comments hold for the point I made about uncertainty and risk taking. The authors should acknowledge this and discuss it.

We now acknowledge the possibility for this alternative explanation in the conclusion section (page 12). As mentioned, in the previous round, we seriously considered this alternative explanation and did additional analyses. Therefore, we now also outline why, based on our results, we believe that this alternative explanation is unlikely to account for what we observed in the experiment (see page 12).

Finally, a plea to the authors to use simpler language. In their replies to my comments, all in the space of a few lines the authors talk about "parochial cooperation", "national discrimination" and then "national parochialism"! These terms might trip off their tongues, but they are almost unreadable to anyone not in the 'ingroup' of this research area. This jargony language extends to the paper. See, for example (I could have chosen many others) lines 146-148: after a particularly heavy paragraph discussing public block, private blocks, indirect reciprocity and national parochialism, we get "This latter finding does not support the hypothesis that national parochialism is a strategy to acquire indirect benefits in a system of reputation-based indirect reciprocity that is contained within a group." What?

We carefully revised the manuscript to make the language simpler and clearer (e.g., page 3, and pp. 9-12). We checked technical terms and revised the paper to make it more accessible to researchers that are not experts of this area. Moreover, when using technical terms, we provide a definition to clarify what we mean (e.g., national parochialism in the abstract). We are grateful for your comment to check into this aspect of our paper. We believe the revised version will be an easier read for our audience.

Reviewer #3 (Remarks to the Author):

I am pleased that throughout this process the authors have taken my comments seriously and addressed the issues I raised.

REVIEWER COMMENTS

Reviewer #3 (Remarks to the Author):

I am pleased that throughout this process the authors have taken my comments seriously and addressed the issues I raised.

We were glad to hear that the reviewer was pleased with the changes made in the last version. We thank the reviewer for the constructive feedback throughout the revision process.